# Deep Learning on Ultrasound Images Visualizes the Femoral Nerve with Good Precision

**DOI:** 10.3390/healthcare11020184

**Published:** 2023-01-07

**Authors:** Johan Berggreen, Anders Johansson, John Jahr, Sebastian Möller, Tomas Jansson

**Affiliations:** 1Biomedical Engineering, Department of Clinical Sciences Lund, Lund University, Lasarettsgatan 37, 22185 Lund, Sweden; 2Intensive and Perioperative Care, Skåne University Hospital, Entregatan 7, 22185 Lund, Sweden; 3Department of Information Technology and Clinical Engineering, Skåne Regional Council, Lasarettsgatan 37, 22185 Lund, Sweden

**Keywords:** artificial intelligence, deep learning, ultrasound, nerve blocks, hip fracture

## Abstract

The number of hip fractures per year worldwide is estimated to reach 6 million by the year 2050. Despite the many advantages of regional blockades when managing pain from such a fracture, these are used to a lesser extent than general analgesia. One reason is that the opportunities for training and obtaining clinical experience in applying nerve blocks can be a challenge in many clinical settings. Ultrasound image guidance based on artificial intelligence may be one way to increase nerve block success rate. We propose an approach using a deep learning semantic segmentation model with U-net architecture to identify the femoral nerve in ultrasound images. The dataset consisted of 1410 ultrasound images that were collected from 48 patients. The images were manually annotated by a clinical professional and a segmentation model was trained. After training the model for 350 epochs, the results were validated with a 10-fold cross-validation. This showed a mean Intersection over Union of 74%, with an interquartile range of 0.66–0.81.

## 1. Introduction

### 1.1. Hip Fracture

Worldwide, the number of hip fractures in 1990 was estimated to be 1.7 million, with an estimated increase to 6 million by 2050. Most patients are elderly, and the most common cause of hip fracture is mild trauma in the home environment. For some patients, hip fractures become a life-threatening condition.

Complications associated with a hip fracture are reported to be high, and mortality within 120 days is 18% [1]. For these patients, the path to the orthopedic ward is long, with many potentially painful transfers while awaiting diagnosis and treatment. According to Bottle and Aylin [2], these steps should be performed as soon as possible, ideally within two hours. Many healthcare organizations have therefore developed a separate track for these patients.

The central and most frequent intervention in the emergency care of patients with suspected hip fracture is pain relief [3]. However, studies indicate that pain relief in this group is in fact insufficient, both pre- and intra-hospitally [4]. Therefore, healthcare professionals who meet these patients in the initial stages have a strategically important role in initiating treatment for acute pain and identifying risk factors for further complications. Inadequate pain relief can, in addition to the acute discomfort, lead to a number of complications. Egbert [5] described over 20 years ago that hospitalized elderly people who experience acute pain have a greater risk of complications, such as atelectasis, pneumonia and embolism.

### 1.2. Advantage of Regional Blockade

Jakopovic et al. [6] have investigated pain management methods in hip fracture patients pre-hospitally. In their study, it emerged that the choice of pain relief was based on preventing procedural pain (e.g., repositioning and transfers) and maintaining pain relief in the chain of care. However, the authors also found that when side effects of central-acting analgesics, such as a drop in blood pressure or respiratory depression, occurred, the staff’s focus changed to treating those instead. The authors also described that commonly selected drugs were ketamine and morphine, which require repeated doses to achieve analgesia through the chain of care. Treatment with central-acting analgesics in this vulnerable group of patients needs to factor in comorbidities and polypharmacy. In addition, titration of central-acting analgesics should be proportional to the patient’s pathology and assessed pain intensity and history [7]. Achieving adequate analgesia with a systemic drug in patients with a suspected hip fracture can therefore be a difficult task.

An alternative would be to use a regional blockade, where the injection of an anesthetic around the nerve innervating the injured area, may provide adequate pain management, without the side effects of central-acting analgesia [8]. Despite the many advantages of regional blockades, these are used in the field to a lesser extent than general analgesia. One reason for this is that general analgesia has a higher probability of success and is more reliable [9]. In patients with hip injuries, adequately identifying the femoral nerve and surrounding anatomical structures is crucial in order to succeed with the femoral block. In addition, poor identification of the anatomical structures can cause damage to the nerve or blood vessels when advancing the needle for injection. However, ultrasound-guided regional blockades have been introduced in environments other than operating wards (e.g., emergency rooms and in disaster areas). In addition, ultrasound can be used to increase the probability of success with a regional blockade. Redborg et al. have shown that in tibial nerve blocks the success rate is 72% with ultrasound and 22% with the traditional landmark-based approach [10]. In addition, Yuan et al. have also shown that the use of ultrasound not only increases the success rate, but also decreases the risk of complications when applying a brachial plexus nerve block [11]. However, ultrasound imaging can be challenging for the inexperienced user. Therefore, appropriate training, both theoretical and practical, is required to succeed with regional blockades to reduce the risk of complications [8].

Opportunities for training and acquiring clinical experience in applying nerve blocks can be a challenge in many clinical settings, decreasing the availability of adequate pain treatments for a patient with a hip fracture. One way to improve patient safety and nerve block success rate can be to incorporate artificial intelligence into the workflow, acting as a real-time ultrasound interpretation tool, thus guiding the inexperienced user when performing the nerve block. We expect this to increase the chances of success with a blockade, reduce interpersonal differences in performance and reduce risks associated with this procedure. In 2019, Huang et al. showed that a model can be trained on low resolution (256 × 256) ultrasound images of the femoral nerve with satisfactory performance, achieving an Intersection over Union (IoU) of 0.656 in a 10-fold cross-validation [12]. However, to apply this technique in a clinical setting, high resolution images need to be used. 

According to Bowness et al., a lot of effort has been put into needle visualization with echogenic materials, but less into the visualization of the anatomy [13]. With newer technical advancements, i.e., AI, this is now a possibility. Furthermore, the authors discuss the need for clinicians to take part in developing AI tools to make nerve blocks accessible to patients under clinicians other than the experienced anesthesiologist. Smistad et al. have used neural networks to highlight nerves and blood vessels. This was done using 49 subjects and an image resolution of 256 × 256. They stated that, “The results are promising; however, there is more work to be done, as the precision and recall are still too low” [14]. Yang et al. managed to achieve better than nonexpert accuracy on segmentation of the interscalene brachial plexus. This was achieved by training a deep neural network on 11,392 images from 1126 patients [15]. The amount of data used for training on ultrasound images can have a large impact. However, collecting and annotating clinical data is expensive and time-consuming, so large datasets are hard to come by. Therefore, in this paper, we propose an approach using a deep learning semantic segmentation model with U-net architecture to identify the femoral nerve in high-resolution ultrasound images to provide support to the inexperienced user when performing a femoral nerve block, with the hypothesis that good precision can be achieved with a more limited dataset of high-resolution images. This model has been developed in a collaboration between engineering and clinical expertise. 

## 2. Materials and Methods

### 2.1. Setting

Ultrasound images were collected from 48 patients undergoing elective orthopedic surgery at Hässleholm Hospital, Sweden. The participants were informed and gave written consent for inclusion in the study. The data collected were images of the femoral nerve and basic demographic data, such as height, weight and age. All data were collected in conjunction with a preoperative meeting with the anesthesiologist.

### 2.2. Ethics Approval and Consent to Participate

Ethics approval was obtained from the Swedish Ethical Review Authority [Dnr: 2020-01835]. The study protocol was approved by the department head. Participation in the study was voluntary; the respondents gave informed consent; all data were kept confidential and participants could withdraw at any time without explanation.

### 2.3. Design

In order to obtain sufficient data to train a deep learning model, images were collected from patients in the clinic. The images were collected from the inguinal crease transversally to the femoral nerve, proximal to the nerve bifurcation, according to Figure 1. 

The patients were randomly divided into three different categories: train, validation and test, with a 70/20/10 percent split. The data were gathered using a Sonosite SII ultrasound system (Bothell, WA, USA) with a HFL38xi/13-6 MHz transducer connected to a laptop. With this setup, short video sequences of the femoral nerve area were recorded by a clinical professional. The video sequences were collected in a 1280 × 720 format at 30 frames per second. We used this format in order to achieve more detailed images, compared to a smaller resolution, which thereby made possible better detection and annotation of the anatomy. From the video sequences, 1410 images were extracted using a script. The training process is illustrated in Figure 2.

The manual annotation of the images was performed in collaboration with an experienced anesthesiologist and the open-source software Labelme, provided by the Computer Science and Artificial Intelligence Lab at the Massachusetts Institute of Technology (https://github.com/CSAILVision/LabelMeAnnotationTool, accessed on 11 May 2021). The images were carefully extracted and annotated only when the femoral nerve was clearly identified.

The segmentation model of choice for this project was U-net, as presented by Ronneberger et al. [16]. U-net has since been widely adopted for biomedical image segmentation tasks, as seen in the study by Guo et al. [17]. U-net is a fully convolutional network, with skip connections giving it the characteristic U architecture. U-net is an encoder-decoder type of network. The encoder combined with down sampling is the first part of the network, and the decoder combined with up sampling is the second part. U-net classifies every pixel in an image, so the input image and output masks have the same resolution.

The model was constructed using Google’s machine learning framework Tensorflow (Google, Mountain View, CA, USA). Basic data augmentation, in the form of rotate, flip and scale, was achieved using the python imgaug library. As an optimizer, the Adam algorithm presented by Kingma et al. set to a learning rate of 0.001 was chosen, and as loss function, categorical cross entropy [18]. The network initialization was performed as described by Ronneberger et al. by drawing the initial weights from a Gaussian distribution with a standard deviation of $\sqrt{\frac{2}{N}}$ [16]. The batch size was set to 64 and the number of epochs to 350. The hardware used to train the model was the Nvidia DGX-2 AI-server (NVIDIA, Santa Clara, CA, USA)

### 2.4. Participants

Collected demographic variables were analyzed using the statistical software SPSS^®^ version 24.0 (SPSS Inc., Chicago, IL, USA). The participants’ height, weight, age and BMI were analyzed using a T-test. Gender differences (proportions) were analyzed with a Chi-square test, and all above-mentioned variables are reported as absolute and relative frequencies, where appropriate.

### 2.5. Model Evaluation

Intersection over Union (IoU) was used to evaluate the precision of the segmentation. This is a method widely adopted for evaluating segmentation. The intersection represents the overlap between the manually annotated image mask and the predicted mask. The union is the combined area of the two masks. Hence, the quota between the two expresses the percentage of overlap of the two total areas. Images from the test category were used for evaluating the model. IoU was used as the primary metric and a comparison was made between the output masks from our trained network and the manual masks annotated by a clinical professional. 

An IoU index value (IV, range: 0–1.0) defined the outcomes. Appellations were then assigned, according to the IV, as follows: 0–0.20 = No or very poor precision; 0.21–0.40 = Poor precision; 0.41–0.60 = Fairly good precision; 0.61–0.80 = Good precision; and 0.81–1.00 = Very good precision [19].

## 3. Results

### 3.1. Participant Demographics

Of the 48 informants, *n* = 25 (52%) were men and *n* = 23 (48%) were women (*p* = 0.083). There were statistically significant differences between the sexes in height (men 1.78 ± 0.75 vs. women 1.66 ± 0.44, *p* = 0.001), weight (men 90 ± 10 vs. women 74 ± 8, *p* = 0.001) and *BMI* (men 28.9 ± 1.8 vs. women 24.6–3.0, *p* = 0.026), but not in age (men 65 ± 10 vs. women 67 ± 11, *p* = 0.434) (Table 1).

### 3.2. IoU Processing

The model was trained on 1410 images with a 70/20/10 split. After training the model for 350 epochs, we achieved a mean IoU of 69%. In order to further validate the result, a 10-fold cross-validation was performed. As suggested by Madani et al. for the performance of a 10-fold crossvalidation, the dataset was split into ten groups, with nine groups being the training set and one being the test set [20]. This process was then executed ten times, iterating through the dataset and creating an average from all ten tests. The result from the cross-validation is displayed in Figure 3. This resulted in a mean IoU of 74% (= Good precision), with an interquartile range of 0.66–0.81 (= Good precision—Very good precision). As a comparison, Figure 4 shows an image with an IoU of 57%.

## 4. Discussion

### 4.1. Study Design

We trained a U-net model using 1410 images from 48 research participants to achieve segmentation of the femoral nerve. This resulted in a mean IoU of 74%, which is comparable to similar studies, such as Huang et al. who achieved an IoU of 64% using 562 images [12].

In this study, we chose to use a segmentation model instead of an object detection model using bounding boxes. The reason for this is the nature of the anatomical structures being detected, since nerves appear as irregular shapes. A bounding box around the anatomical structure (femoral nerve) does not give the delineation needed. If an object detection model could be used instead of segmentation, some benefits regarding inference speed, computational requirements and edge-device deployment could be made.

The population included in this study is similar to the target population for a femoral nerve block. Alpantaki et al. concluded that, by the age of 90, 32% of women and 17% of men will have suffered from a hip fracture. They further conclude that individuals have a two-fold increase in risk of fracture each decade after the age of 50 [21]. The mean age in our study group was 66 years and the gender ratio was 25 men vs. 23 women.

As the aim in capturing the ultrasound images was to create a versatile dataset, the conditions, such as gain, depth and proximal/distal location along the nerve, were varied. This could have increased our model’s versatility at the expense of performance. As with most AI-training tasks, abundant data was essential. Data augmentation is used to enlarge and enrich the dataset and can provide an inoculation against overfitting the model. Negassi et al. state that in many real-world tasks, particularly in medical imaging, only limited amounts of annotated data are available. They also state that insufficient data frequently leads to overfitting [22]. In the present study, we used 1410 images with a 1280 × 720 resolution from 48 participants, resulting in about 29 images from each participant. We chose to collect the data in a 1280 × 720 resolution in order to gain more detail, but this came at the expense of longer model training times and the need for more capable hardware. This was resolved by the use of a Nvidia DGX-2 AI server. When training our model, it started to overfit after 350 epochs using Adam and categorical cross entropy. This is in comparison to Huang et al. (2019), who achieved a similar performance after only 75 epochs [12]. The difference here can probably be explained by the size of the images, dataset and augmentation. However, this was considered an adequate dataset for training, as increasing our dataset with more images sharing similarity with the ones already used did not increase performance. In future studies more data from more individuals could be useful. A drawback of using a larger resolution is the increased inference time. This may have an impact when considering how to deploy the model in a clinical setting, for example, on edge devices with limited computational resources.

### 4.2. Limitations

The annotation technique used may also have impacted our result. We chose to annotate the nerve in as much detail as possible. Another possibility would have been to delineate the area between the fascia iliaca and the iliacus muscle since the actual nerve can be hard to detect. In our case, we only chose images with a visible nerve. We also chose images taken more distally where the nerve had bifurcated, along with images of the single nerve. This probably resulted in a model more attuned to the specific echo from peripheral nerve tissue.

As ground truth we used the annotations from an anesthesiologist with more than 30 years of experience in peripheral nerve blocks. One shortcoming of the result is that it is dependent on the annotations representing our ground truth. Combining annotations from multiple clinical professionals could have reduced this (unknown) variability.

U-net is an architecture specifically designed for biomedical imaging tasks. Nevertheless, visualizing the neural network’s decision process is important in understanding the limitations of the prediction before applying the technique in solving tasks in the clinic. Visualizing on what basis the neurons made their decisions and how they arrived at the final prediction based on our specific dataset could be a challenge. Jimenez-Castaño et al. used GradCam++ for class-activation mapping to reveal relevant learned features for the nerve [23]. Being able to visualize which features of the nerve had the biggest impact for prediction could strengthen the result. Visualization is out of scope for this study, but follow-up studies need to be made to further map the decision process for neural networks on ultrasound images of nerves.

## 5. Conclusions

The approach using a deep learning semantic segmentation model with U-net architecture using 350 epochs showed a mean Intersection over Union of 74%. The results, therefore, indicate that this technique shows potential for assisting the inexperienced user when performing nerve blocks. In further studies, we aim to evaluate what performance is needed for this technique to be used as a clinical tool and at what experience level the tool can provide the most benefit.

## Figures and Tables

**Figure 1 healthcare-11-00184-f001:**
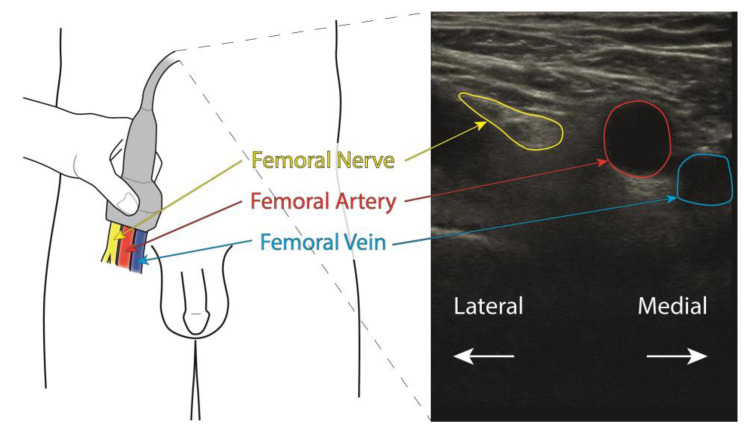
Schematic view of the anatomical situation (left), together with the approximate placement of the ultrasound transducer proximal to the nerve bifurcation. Locations of the femoral nerve, artery and vein are indicated with arrows and text in yellow, red, and blue, respectively. To the right, an example ultrasound image is shown, with the anatomical structures indicated in the same colors.

**Figure 2 healthcare-11-00184-f002:**
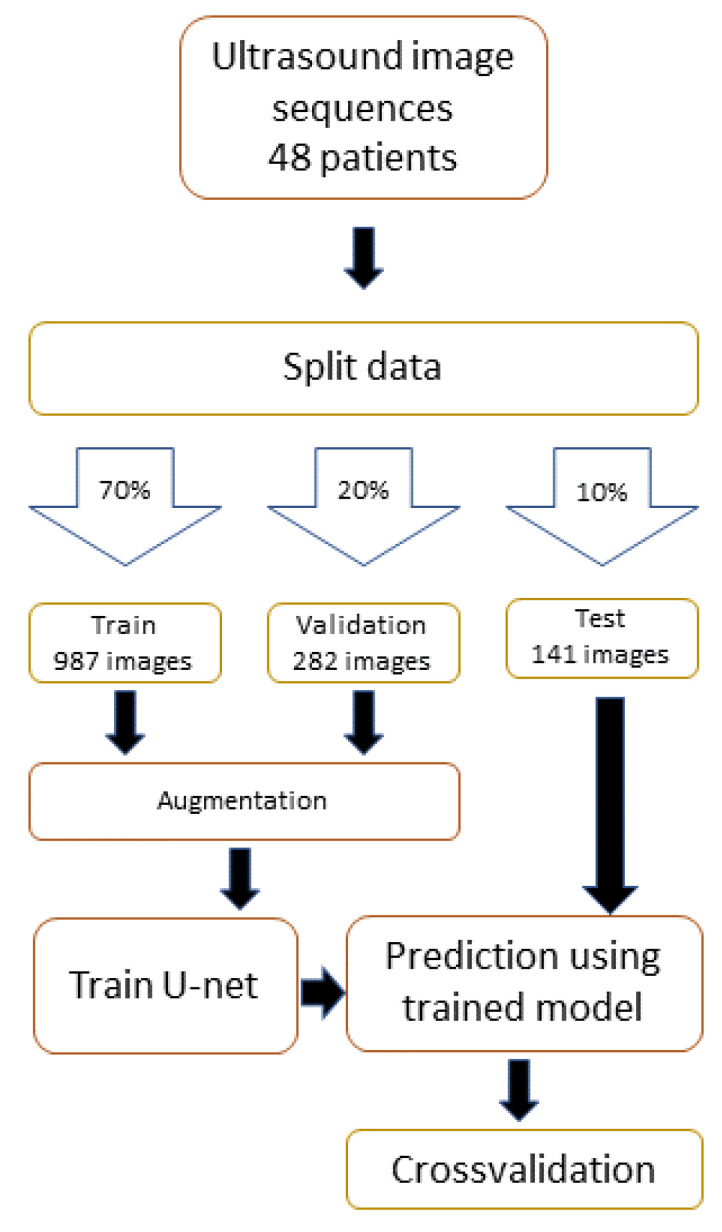
Flow chart of the data-handling and model-training process.

**Figure 3 healthcare-11-00184-f003:**
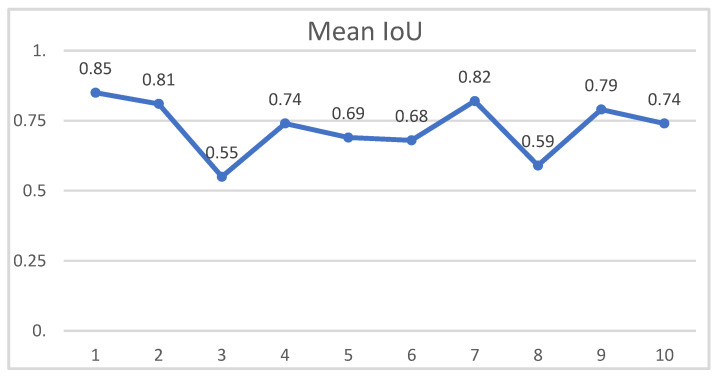
Results of the 10-fold cross-validation. Horizontal axis shows the individual validation sets while the vertical axis shows the correlation for each set.

**Figure 4 healthcare-11-00184-f004:**
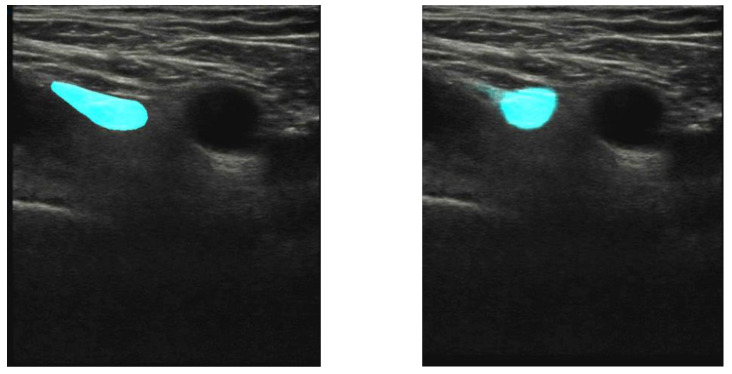
Test image of a participant’s right groin, the same as used in Figure 1. The femoral artery is seen as a black circular area in the center of the image. Left panel: expert’s manual annotation of the femoral nerve (cyan). Right panel: trained U-net segmentation of the femoral nerve. The overlap between the annotated and segmented areas, described as IoU, is 57% in this case.

**Table 1 healthcare-11-00184-t001:** Demographics, absolute and relative frequencies. arithmetic mean [mean ± standard deviation (SD), min-max].

Patients	n = 48	*p*-Value
Gender (men/women)		
n (%)	25 (52)/23 (48)	0.083 *^c^*
Height (mean ± SD, min–max)		
Overall	1.72 ± 0.09, 1.58–1.92	
Men	1.78 ± 0.75, 1.66–1.92	0.001 *^t^* *****
Women	1.66 ± 0.44, 1.58–1.74	
Weight (mean ± SD/min–max)		
Overall	82 ± 13, 58–110	
Men	90 ± 10, 70–110	0.001 *^t^* *****
Women	74 ± 8, 58–88	
Age (mean ± SD/min–max)		
Overall	66 ± 11, 46–88	
Men	65 ± 10, 51–88	0.434 *^t^*
Women	67 ± 11, 46–87	
BMI (mean ± SD/min–max)		
Overall	28.5 ± 2.2, 23.2–30.0	
Men	28.9 ± 1.8, 24.6–30.0	0.026 *^t^* *****
Women	26.8 ± 2.4, 23.2–29.8	

Comparative analysis. ^c^, non-parametric Chi-square test; *^t^*, *t*-test; *****, alpha value *p* ≤ 0.05.

## Data Availability

The datasets generated and/or analyzed during the present study are not publicly available but are available from the corresponding author upon reasonable request. The data have also been added to the data hub of AIDA, a Swedish arena for research and innovation on artificial intelligence for medical image analysis, to be shared with other network partners.

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
