# Peer review of "Deep Learning on Ultrasound Images Visualizes the Femoral Nerve with Good Precision"

_healthcare, 2023, doi:10.3390/healthcare11020184_

Round 1

Reviewer 1 Report

The application of artificial intelligence technology to the ultrasound imaging of the femoral nerve is a useful study with clinical relevance. The methodology is well explained, but the overall impression is that it is too short to be considered for publication, and the usefulness of AI is not convincing enough. AI-based echo-imaging is now widely reported in the field of orthopedics and should be described, including its novelty. My specific suggestions are as follows.

Title

1.  The title should reflect the results, e.g., "Deep learning on ultrasound images visualizes the femoral nerve with high accuracy.

Introduction

1.  The length is appropriate, but the usefulness of ultrasound imaging in femoral nerve delineation and the usefulness of AI technology for ultrasound imaging is less useful. There are few references.

2.  State the hypothesis and novelty with respect to the model used.

3.  Line 82,83

satisfactory performance: Include specific figures and what makes it accurate.

Method

1.  It is well-drawn, but it would be easier to understand if there were diagrams with correct depiction of anatomical location or a flow chart of deep learning.

2.  Please provide a power analysis of the 48 patients.

3.  Was the ultrasound image drawn by a single evaluator?

4.  Do you use image augmentation?

5.  AI technology has a black box problem, and there is a lot of research being done on visualization of the basis for decisions. Are you not considering it this study? It is important to know on what basis the AI is identifying the nerve, and this could have a significant impact on reliability if the needle goes on the US in the block in the future.

Result

1.  Figure 2; Probably quoting images directly from trained models, low resolution. Please describe the anatomical location.

Discussion

1.  In the final two paragraphs, the limitations of the study and the authors' speculation are mixed. Please organize, either by creating a "Limitations" session or by separating the paragraphs.

Author Response

We thank the Reviewers for the constructive comments to our manuscript. We acknowledge their pertinent work and have point-by-point addressed the comments using the “Track Changes” function in the manuscript, also referenced here by line numbers.

Reviewer 1: Comments and Suggestions for Authors

The application of artificial intelligence technology to the ultrasound imaging of the femoral nerve is a useful study with clinical relevance. The methodology is well explained, but the overall impression is that it is too short to be considered for publication, and the usefulness of AI is not convincing enough. AI-based echo-imaging is now widely reported in the field of orthopedics and should be described, including its novelty. My specific suggestions are as follows.

Title

  1. The title should reflect the results, e.g., "Deep learning on ultrasound images visualizes the femoral nerve with high accuracy.

Reply:
In order to better reflect the results, we have now changed the title in the manuscript to “Deep learning in ultrasound images visualizes the femoral nerve with good precision”, using the nomenclature presented in paragraph 2.5 of the manuscript (based on ref 19, Altman 1991).
_________________________________________________________________________
Introduction

  1. The length is appropriate, but the usefulness of ultrasound imaging in femoral nerve delineation and the usefulness of AI technology for ultrasound imaging is less useful. There are few references.

Reply:
More references have been added to the introduction emphasizing the area of deep learning, especially in the field of nerve blocks (lines 75-79). We have also added a more elaborate statement why our study is useful in contrast to earlier work (lines 98-105).

_________________________________________________________________________
2.  State the hypothesis and novelty with respect to the model used.

Reply:
The hypothesis is now clarified and stated in the last paragraph of the introduction (lines 109-110).
_________________________________________________________________________
3.  Line 82,83 satisfactory performance: Include specific figures and what makes it accurate.

Reply:
We have now specified in what way the performance was considered satisfactory, by adding their result for Intersect-over-Union (0.656) to line 92.

_________________________________________________________________________
Method

  1. It is well-drawn, but it would be easier to understand if there were diagrams with correct depiction of anatomical location or a flow chart of deep learning.

Reply:
In order to better contextualize the ultrasound images, an illustration of the anatomical situation, together with the placement of the ultrasound probe, has been added as Fig. 1 (line 134).

_________________________________________________________________________
2.  Please provide a power analysis of the 48 patients.

Reply:
The study is designed as a prospective, non-experimental descriptive evaluation. As the project is designed to be descriptive in nature and no comparisons of subgroups have taken place (e.g., no risk of Type 1-2 errors), there is no performed power calculation for the number of included patients. However, regarding the scope of selected ultrasound images to train a CNN, the literature describes that 500 ultrasound images has previously been deemed sufficient for training. As described in our Method section, our model was trained on 1410 ultrasound images. Therefore, we argue that the number of patients, based on our described inclusion and exclusion criteria, can be considered a representative group and that the number of selected images that were included to train our CNN can be considered adequate based on the objective of our study.

_________________________________________________________________________
3.  Was the ultrasound image drawn by a single evaluator?

Reply:
Yes. We have declared and discussed this in the last paragraph of our section Discussion, lines 277-280:

As ground truth we used the annotations from an anesthesiologist with more than 30 years of experience in peripheral nerve blocks. One shortcoming of the result is that it is dependent on the annotations representing our ground truth. Combining annotations from multiple clinical professionals could have reduced this (unknown) variability.
_________________________________________________________________________
4.  Do you use image augmentation?

Reply:
Yes, we are using basic data augmentation, in the form of rotate, flip and scale using the python imgaug library. This is stated in the last paragraph under 2.3 Design.

_________________________________________________________________________
5.  AI technology has a black box problem, and there is a lot of research being done on visualization of the basis for decisions. Are you not considering it this study? It is important to know on what basis the AI is identifying the nerve, and this could have a significant impact on reliability if the needle goes on the US in the block in the future.

Reply:

The black box problem is ever present in all machine learning tasks and the research regarding visualization is exciting and can bring increased fundamental understanding about the neural network as well as contributing to training optimization and data gathering techniques. Visualization is out of scope for this study, but we added a paragraph addressing this issue, last in the Discussion section (lines 293-302).
_________________________________________________________________________
Result

  1. Figure 2; Probably quoting images directly from trained models, low resolution. Please describe the anatomical location.

Reply:
The image is taken from the test dataset and the prediction made from our trained model. The idea behind showing this image is to give the reader a sense of what a deep learning-based tool can bring to the clinic. The anatomical location is depicted in Figure 1 (line 134).

_________________________________________________________________________
Discussion

In the final two paragraphs, the limitations of the study and the authors' speculation are mixed. Please organize, either by creating a "Limitations" session or by separating the paragraphs.

Reply:
A Limitations session has been added to better structure the content.
_________________________________________________________________________

Again, we thank the reviewers for their valuable comments and we hope that our alterations are in accordance with your criticism, thank you.

Reviewer 2 Report

The authors proposed an approach using a deep learning semantic segmentation model with the U-net architecture to identify the femoral nerve in ultrasound images, and the dataset consisted of 1410 ultrasound images that were collected from 48 patients.

#Comments

1. The details of how to train the U-net in this study need to be clarfied. Such as the loss function, the learning rate, the network initialization?

2. In Page 3, the authors said "Ultrasound images were collected from 48 patients"," The video sequences were then randomly divided to three different categories, train, validation, and test with a 70/20/10 percent split", "From the video sequences 1410 images were extracted and split into train, test and validation categories using a script".

How the authors split the the dataset into train, validation, and test? By 48 patient or by 1410 images? 

If 1410 images are divided into training set, validation set and test set, the images from the same patient will appear in these three datasets. This is a less rigorous way of dividing the data.

3. There are many evaluation metrics to evaluate the segmentation results. Such as Dice and ASD are common metric using for segmentation.

Author Response

We thank the Reviewers for the constructive comments to our manuscript. We acknowledge their pertinent work and have point-by-point addressed the comments using the “Track Changes” function in the manuscript, also referenced here by line numbers.

Reviewer 2: Comments and Suggestions for Authors

The authors proposed an approach using a deep learning semantic segmentation model with the U-net architecture to identify the femoral nerve in ultrasound images, and the dataset consisted of 1410 ultrasound images that were collected from 48 patients.

#Comments

  1. The details of how to train the U-net in this study need to be clarfied. Such as the loss function, the learning rate, the network initialization?

Reply:
Thank you for your comments. We have now added more details regarding the training in the manuscript under 2.3 Design (lines 157-160).

_________________________________________________________________________
2. In Page 3, the authors said, "Ultrasound images were collected from 48 patients"," The video sequences were then randomly divided into three different categories, train, validation, and test with a 70/20/10 percent split", "From the video sequences 1410 images were extracted and split into train, test and validation categories using a script".

How the authors split the dataset into train, validation, and test? By 48 patient or by 1410 images? 

If 1410 images are divided into training set, validation set and test set, the images from the same patient will appear in these three datasets. This is a less rigorous way of dividing the data.

Reply:
This is a very good point and needs to be clarified. The dataset split was done on the patient level and not image level. We needed to make sure that images from a single patient did not “cross-contaminate” in the different categories. This has now been clarified in the manuscript as “The patients were randomly divided to three different categories, train, validation, and test with a 70/20/10 percent split” (lines 133-134).

_________________________________________________________________________
3. There are many evaluation metrics to evaluate the segmentation results. Such as Dice and ASD are common metric using for segmentation.

Reply:
We chose to use IoU since we wanted to have the opportunity to directly relate to other studies on ultrasound that used IoU. IoU is also going to be used in an upcoming study where we evaluate the performance baseline for clinical feasibility, we therefore wanted to achieve a metric consistency.

Again, we thank the reviewers for their valuable comments and we hope that our alterations are in accordance with your criticism, thank you.

Round 2

Reviewer 1 Report

Still not enough figures. A flowchart of deep learning would be easier to understand.

Please add proximal and other locations to the ultrasound image in the figures.

The others are approximate accept.

Author Response

We again thank the Reviewers for the constructive comments to our manuscript. We acknowledge their pertinent work and have point-by-point addressed the comments using the “Track Changes” function in the manuscript, also referenced here by line numbers.

Reviewer 1: Comments and Suggestions for Authors

Still not enough figures. A flowchart of deep learning would be easier to understand.

Please add proximal and other locations to the ultrasound image in the figures.

The others are approximate accept.

Figures

  1. Still not enough figures. A flowchart of deep learning would be easier to understand.

Reply:
A flowchart (Figure 2) has been added to help visualize the data handling and model training process. We have also added a sentence referring to the figure as “The training process is illustrated in Figure 2.” (lines 155-156).

  1. Please add proximal and other locations to the ultrasound image in the figures.

Reply:
We have now added a schematic nerve bifurcation in Figure 1 and added the text in red to the sentence in lines 141: "The images were collected from the inguinal crease transversally to the femoral nerve, proximal to the nerve bifurcation, according to Figure 1." to pinpoint the location of image acquisition more precisely. The legend to Fig 1 now reads "Schematic view of the anatomical situation (left) together with the approximate placement of the ultrasound transducer proximal to the nerve bifurcation. Locations of the femoral nerve, artery and vein are indicated with arrows and text in yellow, red, and blue respectively. To the right an example ultrasound image is shown, with the anatomical structures indicated with the same colors."

We have also added indications of lateral and medial directions in the ultrasound image.

Further, we have noted in the legend of figure 4 that the locations are the same as in figure 1, as the same ultrasound image is used.

Reply:
On a separate note we have performed an additional language check, and have added numerous corrections to the text, marked with “Track changes”.

Again, we thank the reviewers for their valuable comments, and we hope that our alterations are in accordance with your criticism, thank you.

Reviewer 2 Report

All previous comments are properly addressed by the authors.

Author Response

We again thank the Reviewers for the constructive comments to our manuscript. We acknowledge their pertinent work and have point-by-point addressed the comments using the “Track Changes” function in the manuscript, also referenced here by line numbers.

Reply:
We have performed an additional language check, and have added numerous corrections to the text, marked with “Track changes”.

 Again, we thank the reviewers for their valuable comments, and we hope that our alterations are in accordance with your criticism, thank you.